# Rapid Detection of Mycobacterium Tuberculosis Using a Novel Point-of-Care BZ TB/NTM NALF Assay: Integrating LAMP and LFIA Technologies

**DOI:** 10.3390/diagnostics13081497

**Published:** 2023-04-21

**Authors:** Ha Nui Kim, Junmin Lee, Soo-Young Yoon, Woong Sik Jang, Chae Seung Lim

**Affiliations:** 1Department of Laboratory Medicine, Korea University College of Medicine, Seoul 02841, Republic of Korea; 2Department of Laboratory Medicine, Korea University Guro Hospital, Seoul 08308, Republic of Korea; 3Emergency Medicine, College of Medicine, Korea University Guro Hospital, Seoul 08308, Republic of Korea

**Keywords:** tuberculosis, TB, MTB, NALF

## Abstract

Tuberculosis (TB) is one of the leading causes of infectious mortality from a single infectious agent, *Mycobacterium tuberculosis* (MTB). This study evaluated the performance of the newly developed BZ TB/NTM NALF assay, which integrated loop-mediated isothermal amplification and lateral flow immunochromatographic assay technologies, for the detection of MTB. A total of 80 MTB-positive samples and 115 MTB-negative samples were collected, all of which were confirmed by TB real-time PCR (RT-PCR) using either AdvanSure^TM^ TB/NTM RT-PCR Kit or Xpert^®^ MTB/RIF Assay. The performance of the BZ TB/NTM NALF assay was evaluated by calculating its sensitivity, specificity, positive predictive value (PPV), and negative predictive value (NPV) in comparison to those of the RT-PCR methods. Compared to the RT-PCR, the sensitivity, specificity, PPV, and NPV of BZ TB/NTM NALF assay were 98.7%, 99.1%, 98.7%, and 99.1%, respectively. The concordance rate between BZ TB/NTM NALF and RT-PCR was 99.0%. Rapid and simple detection of MTB is essential for global case detection and further elimination of TB. The performance of the BZ TB/NTM NALF Assay is acceptable with a high concordance with RT-PCR, indicating that it is reliable for use in a low-resource environment.

## 1. Introduction

Tuberculosis (TB) is caused by the gram-positive bacillus *Mycobacterium tuberculosis* (MTB) and has been responsible for countless deaths over the centuries. Globally, TB ranks as the thirteenth leading cause of death and surpasses human immunodeficiency virus/acquired immunodeficiency syndrome (HIV/AIDS) as the second-most deadly infectious disease, following coronavirus disease 2019 (COVID-19) [1]. Without treatment, the mortality rate for tuberculosis (TB) is high, reaching approximately 50% [2]. However, with the implementation of currently recommended therapies, which include a 4–6-month course of anti-TB medications, about 85% of patients can achieve a successful recovery [1]. Thus, prompt detection of MTB is essential for accurate TB diagnosis and the initiation of appropriate treatment.

Among the diagnostic methods currently in use, direct microscopic examination of acid-fast-bacilli (AFB) offers a rapid and cost-effective approach, but its limitations include low sensitivity and the inability to differentiate among various mycobacterial species [3]. The culture method, the currently recommended gold standard test for TB, has the disadvantage of a long turnaround time (TAT) of 14–24 days, and is often reported to have decreased sensitivity [4,5,6]. The diagnostic nucleic acid amplification test (NAAT), which includes conventional real-time polymerase chain reaction (RT-PCR) and Xpert MTB/RIF Assay (Cepheid, Sunnyvale, CA, USA) that can simultaneously detect MTB and rifampin (RIF) resistance, is recommended for examining initial respiratory specimens of patients suspected of having TB [4]. The Xpert MTB/RIF Assay, an automated NAAT, has decreased complexity compared with conventional RT-PCR and features a rapid TAT (<2 h) and higher diagnostic rate than culture methods [7,8]. Despite its outstanding analytical abilities, the cost of TB diagnosis has also increased by up to 55%, with the practical limitation for the wide use of Xpert MTB/RIF in underdeveloped areas [8]. Therefore, simpler and less expensive methods can be more appropriate alternatives for diagnosing TB in resource-constrained regions.

According to the World Health Organization (WHO) global TB report in 2021, a reduction in newly diagnosed TB cases has been observed worldwide since the COVID-19 pandemic, mainly due to the disrupted supply and demand for TB diagnostic and treatment services [9]. In settings with a high burden of TB and limited health system capacity, the utilization of point-of-care testing (POCT) can significantly simplify TB detection and treatment, thereby preventing cases from going undiagnosed and untreated.

Among the technologies recommended in the WHO guidelines, loop-mediated isothermal amplification (LAMP) does not require thermal cycling in contrast to conventional PCR [10]. This technology minimizes the TAT of nucleic acid amplification using specially designed primers and DNA polymerase with chain displacement activity [11]. Since the LAMP method requires a constant temperature (60–65 °C), it can be easily used to develop simpler, cheaper, and smaller devices than thermal cyclers [12]. Lateral flow technology, also known as lateral flow immunochromatographic assay (LFIA), is among the most straightforward methods that are widely used for analytical detection in biosensors [13]. LFIA has advantages over other immunoassays in terms of portability, cost-effectiveness, rapid visual readout, and simple sample processing [14]. Furthermore, the LFIA is widely utilized in POCTs due to its ability to accommodate various sample types without requiring additional equipment [13,15]. Nucleic acid lateral flow (NALF) test is an antibody-based immunoassay that uses the principles of both LFIA and LAMP to rapidly and qualitatively detect specifically labeled nucleic acid [16]. The NALF technology can be utilized to develop rapid and user-friendly detection assays that are compatible with molecular methods. Furthermore, the absence of additional equipment requirements in NALF assays renders them ideal for POCT, enabling the full display of their benefits. Various diagnostic tests for rapid MTB detection have been developed using LAMP or LFIA technologies [17,18], and a comprehensive evaluation of their analytical performance in comparison to NAATs is needed. Among the newly developed diagnostic tests, TB-LAMP (Eiken, Tokyo, Japan) and Alere Determine^TM^ TB LAM Ag (Alere Inc., Waltham, MA, USA) have been endorsed by the WHO’s consolidated guidelines for the rapid detection of TB [19]. The amplified product of the TB-LAMP assay, which can be visualized with the naked eye or under ultraviolet light, provides an alternative to smear microscopy in resource-limited settings [20]. The TB LAM Ag test, which detects the lipoarabinomannan (LAM) antigen in urine, serves as a rule-in test for the early identification of TB in patients with HIV-induced immunosuppression [21].

Here, we report the performance of a novel detection assay for TB based on LFIA coupled with LAMP, named the BZ TB/NTM NALF Assay (BioZentech, Seoul, Korea).

## 2. Materials and Methods

### 2.1. Sample Selection and Study Design

Clinical samples were collected from patients suspected of TB between February 2018 and April 2022 at Korea University Guro Hospital. The patients underwent routine screening for TB following the laboratory tests, which included the following: (1) microscopy utilizing auramine-rhodamine fluorescent staining for acid-fast bacilli (AFB) smear, (2) solid and liquid mycobacterial culture utilizing a combination of 3% Ogawa medium (Shinynag, Seoul, Korea) and BACTEC MGIT 960 (Becton Dickinson Microbiology Systems, Sparks, MD, USA), and (3) RT-PCR using either the AdvanSure^TM^ TB/NTM RT-PCR Kit (LG Life Sciences, Seoul, Korea) or Xpert^®^ MTB/RIF Assay (Cepheid Inc., Sunnyvale, CA, USA). This study enrolled 80 MTB-positive samples confirmed by AdvanSure RT-PCR or Xpert MTB/RIF Assay, and 115 samples that tested negative for TB on all of the diagnostic tests described above. The MTB-positive sample consisted of 75 pulmonary (sputum, bronchial washing, and bronchial aspirate) and 5 extrapulmonary (pleural fluid, tissue biopsy, lymph node, and synovial fluid) samples.

Sensitivity, specificity, positive predictive value (PPV), and negative predictive value (NPV) including 95% confidence intervals (CIs) were calculated according to the results of the BZ TB/NTM NALF and RT-PCR results. A statistical analysis was performed using MedCalc^®^ Statistical Software version 20.218 (MedCalc Software Ltd., Ostend, Belgium). This study was approved by the Institutional Review Board of Korea University Guro Hospital (approval number: 2021GR0550).

### 2.2. DNA Extraction and RT-PCR

Sample pretreatment and DNA extraction were performed according to the manufacturer’s instructions. For sputum specimen, an equal volume of sample and 4% NaOH pretreatment solution was mixed and remained room temperature (RT) in 10 min. The mixture with 30 s of vortexing was centrifuged at 4000 rpm for 10 min and 13,000 rpm for 10 min to remove the supernatant. The resulting pellet was washed with 1 mL of phosphate buffer. Nucleic acid extraction from the remaining pellet was performed using the AdvanSure TB/NTM DNA extraction buffer. For bronchial and other specimens, the mixing ratio was adapted based on the specimen’s condition and left at RT for 10 min after mixing. The volume of extraction buffer (50–100 μL) was adjusted based on the amount of the resulting pellet. The Xpert MTB/RIF assay was conducted as per the following protocol: The sample was mixed with 1.5–2 mL of Xpert MTB/RIF solution, with the specific amount adjusted according to the specimen type. The mixture was vortexed and then incubated for 15 min at RT. The resulting mixture was transferred to an Xpert cartridge and loaded into the GeneXpert IV instrument.

The AdvanSure RT-PCR was performed according to the manufacturer’s instruction using SLAN real-time PCR detection system (LG Life Sciences, Seoul, Korea). Targeting both the MTB-specific insertion sequence (IS) *6110* sequences [22] and nontuberculous mycobacteria (NTM)-specific *rpoB* genes [23] enables the AdvanSure RT-PCR to differentiate MTB and NTM simultaneously. The extracted DNA from the specimen was combined with PCR mixtures, primer and probe mix, with positive and negative controls in PCR tubes. A cycle threshold (Ct) value <35 was considered positive for MTB. The Xpert MTB/RIF Assay is an automated nested RT-PCR that amplifies specific sequences of the *rpoB* gene in the MTB complex, allowing for simultaneous qualitative detection of MTB and RIF resistance [24]. A positive MTB result is reported when the Ct values meet the thresholds for at least two probes among the five probes targeting mutations in the RIF-resistance determining region [25].

### 2.3. BZ TB/NTM NALF Assay

For the LAMP process of the BZ TB/NTM NALF Assay, the reagent mixture comprised 14.5 µL of LAMP Mix, 5.5 µL of primers, and 5 µL of the DNA template. The prepared reagent mixture was stored frozen at −20 °C. until the experiment to maintain its stability. The primer sets targeted the IS*6110* sequence of MTB and the *rpoB* gene sequence of NTM. Glyceraldehyde-3-phosphate dehydrogenase was used as the internal control (IC). Each primer set was described in detail previously [26]. Test tubes containing the reagent mixtures were placed on a heat block at 62 °C for 30 min.

For the NALF assay, a detection device with a compact size and plastic housing comprising a sample pad, gold-conjugated pad, nitrocellulose membrane, and absorbent pad was developed (Figure 1). The indirect detection of the LAMP products involves the use of monoclonal antibodies (mAbs) against sets of tags, digoxigenin, fluorescein isothiocyanate (FITC), and biotin (Figure 2A). The LAMP products containing IS*6110* were dual labeled with digoxigenin and biotin, while the products containing *rpoB* comprised FITC and biotin. The dual-labeled amplicons are shown in Figure 2B,C. Discriminative mAbs against each tag were sprayed onto nitrocellulose membranes, resulting in T1, T2, and IC test lines. When the mAbs bind to their respective tags, streptavidin gold conjugates bind to the biotin present in the dual-labeled amplicons.

### 2.4. Interpretation of BZ TB/NTM NALF Assay Results

After the LAMP procedure, 180–200 µL of buffer was applied to the test tube and mixed gently. The mixture was then dispensed onto the sample pad and flowed along the nitrocellulose membrane towards each test line (T1, T2, and IC). The resulting complexes were immobilized on the strip at their corresponding test lines. The presence or absence of red lines was visually assessed after 10 min.

The test device has three lines, with the first line corresponding to *IS6110* and indicating a positive result for MTB, the second line corresponding to *rpoB* and indicating a positive result for NTM, and the third line corresponding to the IC (Figure 2D). The control line must always be present in all tests for the results to be valid. The absence of the control line with or without the presence of other test lines should be interpreted as invalid and be retested.

## 3. Results

The mean TAT for the BZ TB/NTM NALF Assay was 40 min, while the TAT for both RT-PCR was 2 h on average, respectively.

The results of BZ TB/NTM NALF Assay were compared to those of RT-PCR. One false positive and one false negative result were noted (Table 1). The sensitivity, specificity, PPV, and NPV of the BZ TB/NTM NALF Assay were 98.7% (95% CI: 93.2–99.9%), 99.1% (95.2–99.9%), 98.7% (91.8–99.8%), and 99.1% (94.2–99.8%), respectively. The concordance rate between BZ TB/NTM NALF and RT-PCR was 99.0% (193/195).

The final culture results obtained after six weeks were compared to those of RT-PCR and BZ TB/NTM NALF. A total of 12 discrepant cases were identified, which tested positive on both RT-PCR and BZ TB/NTM NALF but negative on culture. The characteristics of these cases were summarized in Table 2. All three cases that were positive for Xpert but negative for culture had low Xpert results. Additionally, four out of the five extrapulmonary samples showed negative results in mycobacterial culture. All patients were clinically diagnosed with MTB infection, which was supported by relevant diagnostic tests including chest X-ray (CXR), chest computed tomography (CT), or biopsy. The majority of patients received anti-TB medication, except for three cases in which treatment was not initiated due to concurrent conditions such as cancer, pneumonia, and poor general condition. Anti-TB treatment included isoniazid, rifampicin, pyrazinamide, and ethambutol.

The TB-LAMP and Alere LAM assays, two WHO-endorsed rapid diagnostic tests other than RT-PCR and Xpert MTB/RIF Assay, were compared with the BZ TB/NTM NALF concerning their principles, readout, turnaround time, performance, and per-test cost (Table 3). The amplified turbid, fluorescent product of the TB-LAMP assay can be visually observed with the naked eye under ultraviolet (UV) light. The results of the Alere LAM assay can be examined with the naked eye for visible bands, similar to the BZ TB/NTM NALF assay.

## 4. Discussion

TB is one of the leading causes of illness worldwide and is responsible for significant numbers of deaths annually [4,10]. As a single infectious pathogen, TB was the formal leading cause of global death before the COVID-19 pandemic. Although TB is a curable and controllable disease through an established drug regimen, adequate medical intervention is sometimes hampered by several external factors such as poverty, nutrient deficiencies, and concurrent infection. Incomplete reports on the incidence of TB have also been a barrier to meeting the WHO goal of its eradication by 2035 [28].

According to the 2021 global TB report by the WHO [9], a significant global decline in the incidence of newly diagnosed cases of TB has been reported. The main contributions of these reductions were noted among countries with the highest TB burden. The reason for these global decreases between 2019 and 2020 was explained not by a true decrease in the incidence of TB, but by a decrease in the need for TB diagnosis and treatment owing to the disrupted healthcare system caused by the COVID-19. Restrictions on movement due to lockdown, tendency to avoid using hospital facilities due to concerns about infection, and confusion resulting from the similarities in respiratory symptoms were cited as possible reasons for disruptions in the TB monitoring system. The widening gap between the actual incidence of TB infection and its reported incidence of TB in 2020 versus 2019 suggests an increased number of patients with delayed TB detection and treatment. The number of deaths caused by TB in 2020 increased worldwide up to 1.5 million for the first time in over a decade, regardless of region and country [29]. The disruption of health services is more evident in low- to middle-income countries, and deaths from TB could increase by up to 20% according to a high-burden modeling analysis [30].

The rapid molecular test, the Xpert MTB/RIF Assay, was endorsed by the WHO in 2010 with the recommendation of its use as the initial diagnostic test in cases of suspected TB [31]. The Xpert has the advantage of detecting RIF resistance, as well as high sensitivity and specificity in comparison with the AFB microscopy [32]. Compared with the AdvanSure TB/NTM RT-PCR, the Xpert provided better sensitivity, especially in AFB smear–negative cases [33,34]. However, the actual use of the Xpert is limited in high-burden countries with limited resources. The attempt to implement the Xpert MTB/RIF assay in Uganda demonstrated practical challenges in a real-world low-income settings, with no increase in the initiation of TB treatment over AFB smear microscopy [35]. In addition, analytical errors, cartridge stockout problems, lack of data connections for the transmission of test results, and management issues such as repair make it difficult to implement the Xpert MTB/RIF Assay in environments unsuitable for routine laboratory workups [35,36].

The current study evaluated the performance of the BZ TB/NTM NALF Assay, a device that can be a feasible alternative in resource-limited environments. The BZ TB/NTM NALF Assay combines the principles of LAMP and LFIA. The LAMP assay has potential advantages over traditional PCR due to its ability to provide freedom from the need for specialized PCR equipment and trained specialists [37]. The reduced power consumption and easy adaptation to the POCT platform are among its merits [38]. In addition, the simple and rapid LFIA assay has expanded its applications to various fields where rapid tests are required, owing to its low development costs and ease of production [39]. The BZ TB/NTM NALF Assay uses LAMP to amplify the DNA of MTB and LFIA to detect the amplified DNA, providing a rapid and easy-to-use detection method for TB. Another problems have been reported during use of the Xpert MTB/RIF assay in Mozambique in addition to testing error rates including difficulty using English-language software [36]. With the BZ TB/NTM NALF Assay, on the other hand, the results can be interpreted by simple visual reading with the naked eye without the need for another reading device. The simplicity and ease of use of the BZ TB/NTM NALF Assay can help overcome technical and logistical challenges often encountered when implementing more complex assays such as the Xpert MTB/RIF assay in real-world settings. Compared to other WHO-endorsed rapid diagnostic tests, the BZ TB/NTM NALF Assay exhibited a performance that was either compatible with or superior to the TB-LAMP, offering a reduced TAT and eliminating the need for UV light usage in readout. Additionally, the sensitivity of the Alere LAM assay is reported to be low in immunocompetent adults, with a rate of 14.0% (95% CI: 4.0–38.0%), rendering it unsuitable as a general diagnostic test for TB [40]. Consequently, the BZ TB/NTM NALF Assay showed the advantages of easy readout, rapid results, and cost-effectiveness.

One of the limitations of this study is that the reference method used to evaluate the performance of the BZ TB/NTM NALF assay was RT-PCR, rather than the mycobacterial culture method which is considered the gold standard for TB diagnosis. Although the mycobacterial culture method is considered the gold standard for detecting MTB [1], it is time consuming and can take up to six weeks to confirm the presence of TB. This delay in diagnosis is not suitable for prompt initiation of treatment in patients suspected of having active TB.

Out of the 12 discrepant cases in our study, 4 cases were also positive for MTB using RT-PCR in biopsy specimens or exhibited consistent histological findings for TB infection. In the remaining discrepant cases, other diagnostic evidence including CXR or CT features were suggestive of active TB. In cases where MTB culture results were negative, a clinical diagnosis can be made based on consensus, considering factors such as clinical and radiological presentation, hematological or histological findings, and response to anti-TB treatment [41]. Of the 12 cases with discrepancies between Xpert and culture results, 1 case with a low Xpert result (case number 9 in Table 2) was later found to be culture positive in a repeat test conducted one month later. Thus, a negative MTB culture alone may not be sufficient to rule out TB, especially in patients with clinical and radiological features that suggest active disease.

Several possible reasons may account for positive RT-PCR results but negative culture results. Due to its ability to detect both viable and non-viable DNA, molecular methods such as RT-PCR may yield positive results for MTB that do not necessarily indicate the presence of viable bacilli, in contrast to the culture method which specifically detects viable bacilli [42]. The discrepancy may be attributed to reduced viability of the bacilli due to inappropriate laboratory procedures used during the handling of respiratory specimens. The reagents used for decontamination, centrifugation, and storage can have a negative impact on the viability of the bacilli if not properly performed [43,44]. Given that all Xpert-positive and culture-negative cases exhibited low to very low Xpert results, the low count of bacilli may have contributed to the negative culture result [45].

It is worth noting that the BZ TB/NTM NALF Assay yielded positive results consistent with the RT-PCR in all 12 cases where discrepancies were observed between RT-PCR and culture results. While the clinical impact of culture negative and RT-PCR positive cases may be subject to debate, the BZ TB/NTM NALF Assay demonstrated the ability to detect both viable and non-viable MTB DNA, showing sensitivity comparable to RT-PCR.

However, our study utilized a case—control design with TB RT-PCR as the reference method, which carries a risk of overestimating diagnostic accuracy due to the potential for spectrum bias. A prospective blind comparison of the diagnostic and reference test methods in a consecutive series of patients from a relevant population is generally considered the optimal study design for assessing test accuracy [46]. The case–control study design can affect the estimation of diagnostic accuracy, leading to an overestimation of sensitivity and specificity [47].

Another limitation of this study is the absence of evaluation for NTM detection in the performance assessment of the BZ TB/NTM NALF Assay. This was due to the small number of NTM-positive samples available during the sample collection period. Given the importance of accurate differentiation between MTB and NTM for early TB treatment, future research is needed to evaluate the performance of the BZ TB/NTM NALF Assay for differentiating between MTB and NTM.

## 5. Conclusions

The analytical performance of the BZ TB/NTM NALF Assay was excellent compared to that of the TB RT-PCR, with a sensitivity and specificity of 98.0% and 98.8%, respectively. Our findings suggest that the LAMP and LFIA methods can be alternative diagnostic tests over NAATs in developing countries because they do not require expensive equipment or skilled personnel and are cost effective and rapid. These tests have proven wide availability, faster TAT, convenient use, and easy interpretation of test results. The BZ TB/NTM NALF Assay, which consists of a combination of LAMP and LFIA, showed excellent performance compared to that of the TB RT-PCR, AdvanSure TB/NTM RT-PCR kit or Xpert MTB/RIF Assay. The distinctive advantages of the BZ TB/NTM NALF Assay over NAAT make it useful in low-resource settings and will aid in the recovery of the disturbed TB monitoring system after the COVID-19 pandemic.

## Figures and Tables

**Figure 1 diagnostics-13-01497-f001:**
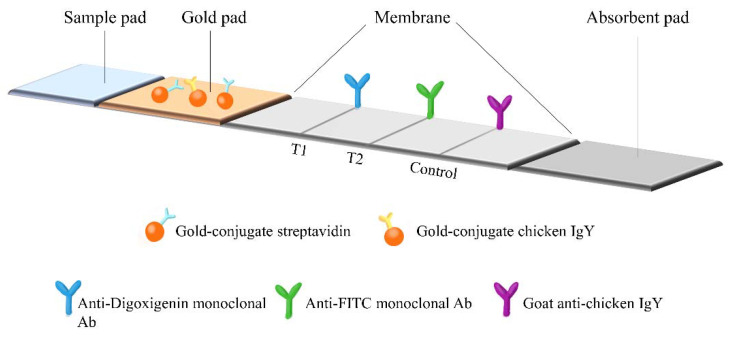
Schematic representation for the principle of lateral flow immunochromatographic assay in BZ TB/NTM NALF Assay.

**Figure 2 diagnostics-13-01497-f002:**
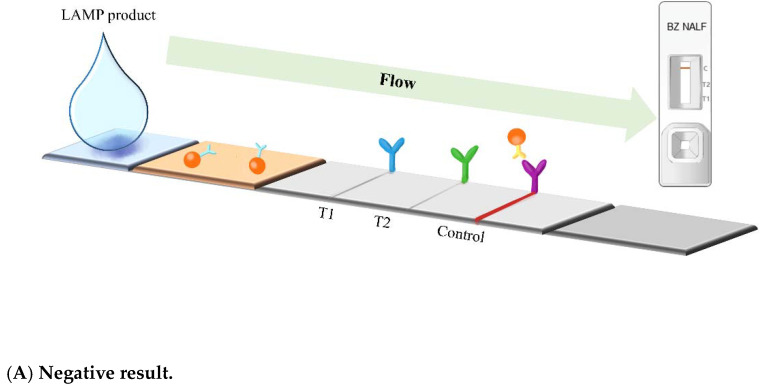
Interpretations of BZ TB/NTM NALF Assay with schematic illustrations of corresponding test cassette (upper right). A control line should be present in all tests and the absence of this line is considered an invalid result. (**A**) Negative result, one red line appears on control region. (**B**) MTB-positive result, two red lines present on T1/T2 and control. (**C**) NTM-positive result, two red lines appear on T2 and control region. (**D**) A photograph of test device representing MTB-positive, NTM-positive, and negative results (**left** to **right**).

**Table 1 diagnostics-13-01497-t001:** Diagnostic performance of BZ TB/NTM NALF assay compared to RT-PCR (AdvanSure^TM^ TB/NTM RT-PCR Kit or Xpert^®^ MTB/RIF assay). The results of mycobacterial culture method were also compared to the TB RT-PCR.

Type of Assay		RT-PCR		Performance (95% Confidence Interval, %)			
		Positive	Negative	Sensitivity	Specificity	PPV	NPV
BZ TB/NTMNALF Assay	Positive	79	1	98.7% (93.2–99.9%)	99.1% (95.2–99.9%)	98.7% (91.8–99.8%)	99.1% (94.2–99.8%)
	Negative	1	114				
Culture	Positive	68	0	85.0% (75.3–92.0%)	100% (96.8–100%)	100%	90.5% (85.0–94.2%)
	Negative	12	115				

PPV: positive prediction value, NPV: negative prediction value.

**Table 2 diagnostics-13-01497-t002:** Characteristics of discrepant results between TB PCR either by AdvanSure^TM^ TB/NTM RT-PCR Kit or Xpert^®^ MTB/RIF assay and mycobacterial culture. Patient information including sex, age, clinical diagnosis, anti-TB treatment status, and relevant findings from chest X-ray (CXR), computed tomography (CT), or biopsy, as well as the type of TB RT-PCR and tested specimen, were summarized.

	Sex	Age	TB PCR Result	Specimen	Clinical Diagnosis	Anti-TB Treatment	CXR, CT or Biopsy
1	M	82	AdvanSure	Pleural fluid	TB pleurisy	Performed	CT: diffuse pleural thickening, calcification, and large effusion, r/o TB pleurisy
2	M	76	AdvanSure	Sputum	Pneumonia with TB reactivation	Not performed due to poor condition	CT: multiple small nodules with consolidations, r/o active pulmonary TB
3	M	45	AdvanSure	Sputum	Pulmonary TB	Performed	Biopsy: positive for MTB RT-PCR in lung tissue
4	M	74	Xpert (very low)	Bronchial washing	Non-small cell lung cancer with TB	Not performed due to chemotherapy	CT: pulmonary TB with bronchiectasis
5	F	24	AdvanSure	Neck mass	TB cervical lymphadenopathy	Performed	Biopsy: chronic granulomatous inflammation with caseation necrosis, consistent with TB, in neck tissue
6	F	73	AdvanSure	Sputum	Adenocarcinoma with skull metastasis, r/o Miliary TB	Not performed due to poor condition	CT: patchy opacity and tiny nodular opacities in both lungs, r/o pneumonia and metastasis
7	F	88	AdvanSure	Fine needle aspirate	TB spondylitis	Performed	CT: Subsegmental atelectasis
8	M	57	AdvanSure	Swab	TB epididymitis	Performed	CXR: calcific granulomas in both lungsBiopsy: chronic caseating granulomatous inflammation, consistent with TB in epididymis
9	M	46	Xpert (low)	Bronchial washing	Multi-drug resistant TB	Performed	CXR: cavitary lesion, active pulmonary TB.
10	M	37	Xpert (very low)	Bronchial washing	Pulmonary TB	Performed	CT: peribronchial consolidation and ground-glass opacities (GGO)
11	M	27	AdvanSure	Sputum	TB cervical lymphadenopathy	Performed	CXR: pulmonary TB in lung, activity undetermined Biopsy: positive for MTB RT-PCR in lymph node
12	F	74	AdvanSure	Sputum	TB pleurisy	Performed	CT: patchy GGOs and consolidations

**Table 3 diagnostics-13-01497-t003:** Comparison of WHO-endorsed TB diagnostic tests: TB-LAMP and Alere DetermineTM TB LAM Ag, incorporated loop-mediated isothermal amplification (LAMP) or lateral flow technology, vs. BZ TB/NTM NALF assay.

	TB-LAMP (Eiken) [19,27]	Alere Determine^TM^ TB LAM Ag (Alere) [19]	BZ TB/NTM NALF Assay
Test principle	LAMP reaction	Lateral flow technology	Nucleic acid lateral flow (LAMP and lateral flow technology)
Readout	Under ultraviolet light with naked eye	Naked eye	Naked eye
Turnaround time	~1 h	~25 min	~40 min
Sensitivity (95% confidence interval, %)	80.9% (76.0–85.1%)	Symptomatic participants: 42.0% (31.0–55.0%) Unselected participants: 35.0% (22.0–50.0%)	98.7% (93.2–99.9%)
Specificity (95% confidence interval, %)	96.5% (94.7–97.7%)	Symptomatic participants: 91.0% (85.0–95.0%) Unselected participants: 95.0% (89.0–98.0%)	98.7% (91.8–99.8%),
Weighted average per-test cost	US$13.78–16.22	US$3.5	US$4.0
Recommendation in WHO	Replacement for sputum smear microscopy in adults with signs and symptoms of TB	Assist diagnosis of active TB in HIV-positive adults, adolescents and children:	

## Data Availability

The data supporting the findings of this study are available from the corresponding author upon reasonable request.

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
