# Peer review of "Rapid Detection of Mycobacterium Tuberculosis Using a Novel Point-of-Care BZ TB/NTM NALF Assay: Integrating LAMP and LFIA Technologies"

_diagnostics, 2023, doi:10.3390/diagnostics13081497_

Round 1

Reviewer 1 Report

This work introduces an interesting platform for detection of mycobacterium tuberculosis, which suits well for point-of-care diagnostics. 

I have one comment: I suggest the author to add a table to make a detailed comparison between their technology/devices with the available products/technologies on the market, including the performance, cost, time, etc.

Author Response

Response to Reviewer 1 Comments

Thank you for giving me the opportunity to submit a revised draft of my manuscript to Diagnostics. We appreciate the time and effort that the reviewers have dedicated to providing your valuable feedback on my manuscript. The modified sentences were highlighted in the manuscript.

Point 1: This work introduces an interesting platform for detection of mycobacterium tuberculosis, which suits well for point-of-care diagnostics.

I have one comment: I suggest the author to add a table to make a detailed comparison between their technology/devices with the available products/technologies on the market, including the performance, cost, time, etc.

Response 1: Thank you for your insightful comment. We have taken your suggestion into account and incorporated your valuable feedback by adding Table 3, which compares our technology with the WHO-endorsed rapid diagnostic tests, namely TB-LAMP (Eiken, Tokyo, Japan) and Alere DetermineTM TB LAM Ag (Alere Inc., Waltham, MA, USA). We believe this addition will provide readers with a better understanding of the relative merits of our platform in comparison to these existing diagnostic methods.

Reviewer 2 Report

In this manuscript, authors presented a rapid and user-friendly detection assay by combining the LAMP and LFIA techniques for TB diagnosis. This NALF-based approach offered simple visual reading to interpret the detection results, from the clinical samples. The sensitivity, specificity, PPV, and NPV were fully listed by comparing with the culture method and RT-PCR, to highlight the performance of the proposed device. Overall, the tested results are convincible, and sufficient information was mentioned to describe the limitation of proposed device in this study. It is easy to read this well-written and structured manuscript. I would suggest it for the publication.

Author Response

Response to Reviewer 2 Comments

Thank you for giving me the opportunity to submit a revised draft of my manuscript to Diagnostics. We appreciate the time and effort that the reviewers have dedicated to providing your valuable feedback on my manuscript.

Point 1: In this manuscript, authors presented a rapid and user-friendly detection assay by combining the LAMP and LFIA techniques for TB diagnosis. This NALF-based approach offered simple visual reading to interpret the detection results, from the clinical samples. The sensitivity, specificity, PPV, and NPV were fully listed by comparing with the culture method and RT-PCR, to highlight the performance of the proposed device. Overall, the tested results are convincible, and sufficient information was mentioned to describe the limitation of proposed device in this study. It is easy to read this well-written and structured manuscript. I would suggest it for the publication.

Response 1: We appreciate your thoughtful review and positive feedback on our manuscript. We are grateful for your acknowledgment of our manuscript's well-written and structured nature, as well as your recommendation for publication.
